

# Correlation and agreement of regional cerebral oxygen saturation measured from sensor sites at frontal and temporal areas in adult patients undergoing cardiovascular anesthesia

Sirirat Tribuddharat, Kriangsak Ngamsaengsirisup, Phatcharakamon Mahothorn and Thepakorn Sathitkarnmanee

Department of Anesthesiology, Faculty of Medicine, Khon Kaen University, Khon Kaen, Thailand

Corresponding author
Thepakorn Sathitkarnmanee,
thepakorns@gmail.com

## ABSTRACT

**Background:** The function and viability of the brain depend on adequate oxygen supply. A decrease in cerebral blood supply causing cerebral desaturation may lead to many neurological complications. Direct measurement of regional cerebral oxygen saturation ($rScO_2$) assists in early detection and management. Near-infrared spectroscopy (NIRS) has been introduced for measuring $rScO_2$. A pair of sensors are attached to the right and left forehead. However, there are some situations where the forehead of the patient is not accessible for sensor attachment (*e.g.*, neurosurgery involving the frontal area; a bispectral index (BIS) sensor already attached, or a wound to the forehead); therefore, alternate sites for sensor attachment are required. The temporal area was proposed as an alternate site. The objective of this study was to assess the correlation and agreement of $rScO_2$ measured at the forehead *vs.* the temporal area.

**Methods:** Adult patients undergoing cardiothoracic or vascular surgery were monitored for $rScO_2$ using two pairs of ForeSight sensors. The first pair (A1 and A2) were attached to the right and left forehead, while the second pair (B1 and B2) were attached to the right and left temporal area. The $rScO_2$ values measured from A1 *vs.* B1 and A2 *vs.* B2 were assessed for correlation and agreement using the Bland-Altman analysis.

**Results:** Data from 19 patients with 14,364 sets of data were analyzed. The data from A1 *vs.* B1 and A2 *vs.* B2 showed moderate positive correlation (r = 0.627; $P < 0.0001$ and r = 0.548; $P < 0.0001$). The biases of A1 *vs.* B1 and A2 *vs.* B2 were −2.3% (95% CI [−2.5 to −2.2]; $P < 0.0001$) and 0.7% (95% CI [0.6–0.8]; $P < 0.0001$). The lower and upper limits of agreement of A1 *vs.* B1 were −17.5% (95% CI [−17.7 to −17.3]) and 12.8% (95% CI [12.6–13.0]). The lower and upper limits of agreement of A2 *vs.* B2 were −14.6% (95% CI [−14.8 to −14.4]) and 16.0% (95% [CI 15.8–16.3]).

**Conclusions:** The $rScO_2$ values measured from sensors at the frontal and temporal areas show a moderate correlation with sufficiently good agreement. The temporal area may be an alternative to the frontal area for cerebral oximetry monitoring.

## INTRODUCTION

The brain's normal function and viability depend on an adequate oxygen supply. The current methods to assess oxygen delivery to the brain include measuring blood pressure and arterial oxygenation *via* pulse oximetry or blood gas analysis which are surrogate parameters (*Pollard et al., 1996*). Near-infrared spectroscopy (NIRS), a non-invasive cerebral oximetry monitor, has been introduced to directly measure regional cerebral oxygen saturation ($rScO_2$) (*Wahr et al., 1996*; *Owen-Reece et al., 1999*; *Vegh, 2016*; *Hogue et al., 2021*). The light in the near-infrared spectrum can penetrate tissue containing bone and soft tissue/gray matter up to 2.5 cm in depth. Sensors are placed at fixed distances from a light emitter, and algorithms subtract superficial from deep light absorption to provide an index of tissue oxygenation (*Steppan & Hogue, 2014*).

Cardiac surgery may reduce cerebral perfusion due to a decrease in cardiac output and blood pressure leading to cerebral desaturation; thus, it is associated with many neurological complications (*Hogue, Palin & Arrowsmith, 2006*; *Gottesman, McKhann & Hogue, 2008*; *Selnes et al., 2012*; *Krause et al., 2020*). It is the most common procedure benefitting from NIRS, which has resulted in lower rates of postoperative stroke, postoperative cognitive dysfunction (POCD), and postoperative delirium (POD) in adult cardiac surgery (*Zheng et al., 2013*). NIRS measurement does not rely on pulsatile flow as does pulse oximetry; rather, it averages the oxygenation of the artery, capillary, and venous flow of the underlying tissue. There is still no consensus on the criteria for cerebral desaturation, but the commonly used criterion is a greater than 20% reduction of $rScO_2$ from baseline or an absolute value of less than 50%, although a reduction greater than 10% was set as the threshold for early intervention (*Murkin & Arango, 2009*; *Deschamps et al., 2016*). Since there is broad steady state variability among individuals and wide dynamic error in measurements, cerebral oximetry should be interpreted as a trend rather than an absolute $rScO_2$ value (*Henson et al., 1998*; *Bickler, Feiner & Rollins, 2013*). A network meta-analysis revealed that by maintaining a $rScO_2 > 80\%$ of baseline, NIRS was associated with protection against POCD/POD in cardiac surgery (pooled odds ratio 0.34; 95% CI [0.14–0.85]) (*Ortega-Loubon et al., 2019*).

In order to perform cerebral oximetry monitoring, a pair of adhesive sensors are attached to the right and left sides of the forehead of the patient to measure the $rScO_2$ of the frontal cortex (*Hogue et al., 2021*). The frontal cortex receives blood supply from two branches of the internal carotid artery: the anterior and the middle cerebral artery (*El-Baba & Schury, 2022*). There are some situations where the forehead of the patient is not accessible for sensor attachment (*e.g.*, neurosurgery involving the frontal area, a bispectral index (BIS) sensor already attached, or a wound to the forehead); therefore, alternate sites for sensor attachment are required. The temporal area was proposed as an alternative site for sensor attachment for measuring temporal lobe $rScO_2$. The temporal lobe forms the

cerebral cortex along with the occipital lobe, the parietal lobe, and the frontal lobe. The temporal lobe receives its blood supply from the internal carotid system and the vertebrobasilar artery (*Patel, Biso & Fowler, 2022*). Furthermore, the structures covering frontal and temporal lobes—the skin, subcutaneous fat, thin layer of muscle, and skull— are comparable; thus, the $rScO_2$ measured from the forehead and temporal area should provide comparable information.

The objective of this study was to assess the correlation and agreement of $rScO_2$ as measured using sensors attached to the forehead *vs.* the temporal area. The proposed criterion for an acceptable agreement was a bias of less than 5%, with the lower and upper limits of agreement being within 15% of the bias.

## METHODS

The study was approved on July 30, 2021 by the Khon Kaen University Ethics Committee for Human Research (HE641311). The study was registered on October 9, 2021 at ClinicalTrials.com (NCT05087836). The study was performed as per the Declaration of Helsinki and the ICH GCP, and all participants gave written informed consent before being recruited.

This was a prospective descriptive study. The sample size of 21 patients was based on the expected correlation coefficient of 0.6, an α-value of 0.05, and a β-value of 0.2, with a 20% drop-out. The inclusion criteria were patients (1) of any sex; (2) age 18 or older; (3) undergoing elective cardiothoracic or vascular surgery at Srinagarind Hospital or Queen Sirikit Heart Center of the Northeast, Khon Kaen University, Khon Kaen, Thailand; and, (4) with an American Society of Anesthesiologists (ASA) classification II to III. The exclusion criteria were patients with (1) intra-cranial or carotid vascular disease; (2) previous surgery of face or brain; (3) abnormal anatomy or fibrosis of face; and/or (4) redo surgery.

All patients received standard cardiac anesthesia care. The intraoperative monitoring consisted of electrocardiogram, pulse oximetry, non-invasive blood pressure, temperature, capnography, anesthetic gas analyzer, and urine output. In addition, the $rScO_2$ was monitored using ForeSight sensors and a HemoSphere monitor (Edwards Lifesciences, Irvine, CA, USA). Each patient was monitored with two pairs of ForeSight sensors connected to the same HemoSphere monitor. The first pair (A1 and A2) were attached to the right and left forehead area, while the second pair (B1 and B2) were attached to the right and left temporal region of the patient (Fig. 1). The proximal light source of the temporal sensor was located between the eyebrow's tail and the helix root outside the hairline, while the distal end of the deep detector was located upward below the hairline (Fig. 2). The sensors were then secured by covering with Tegaderm films (3 M, Minn, Saint Paul, MN, USA). The $rScO_2$ values from all sensors were recorded every 20 s. After the operation, the data from all sensors were downloaded for analysis.

All patients received fentanyl 2–3 $\mu g \cdot kg^{-1}$ and midazolam 1–2 mg as premedication. Anesthesia was induced with propofol 2–3 $mg \cdot kg^{-1}$ or etomidate 0.3 $mg \cdot kg^{-1}$. Endotracheal intubation was facilitated with cis-atracurium 0.2 $mg \cdot kg^{-1}$. The ventilation was controlled using a tidal volume of 8 $mL \cdot kg^{-1}$ and a rate of 12–14 $breaths \cdot min^{-1}$,

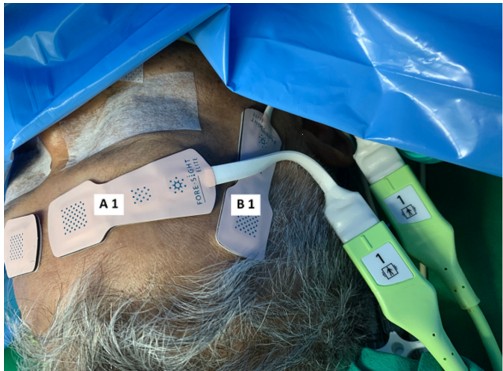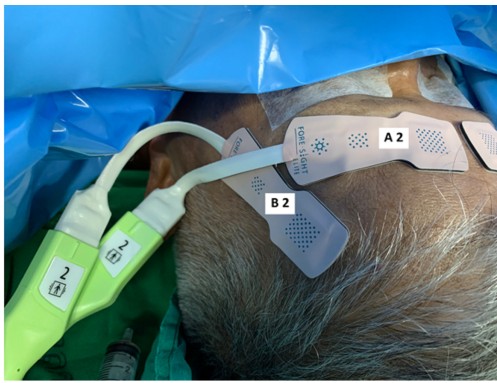

**Figure 1 Locations of two pairs of ForeSight sensors.** (Left) A1 and A2 are the locations for the first pair of sensors on the right and left forehead areas. (Right) B1 and B2 are the locations for the second pair of sensors on the right and left temporal areas.

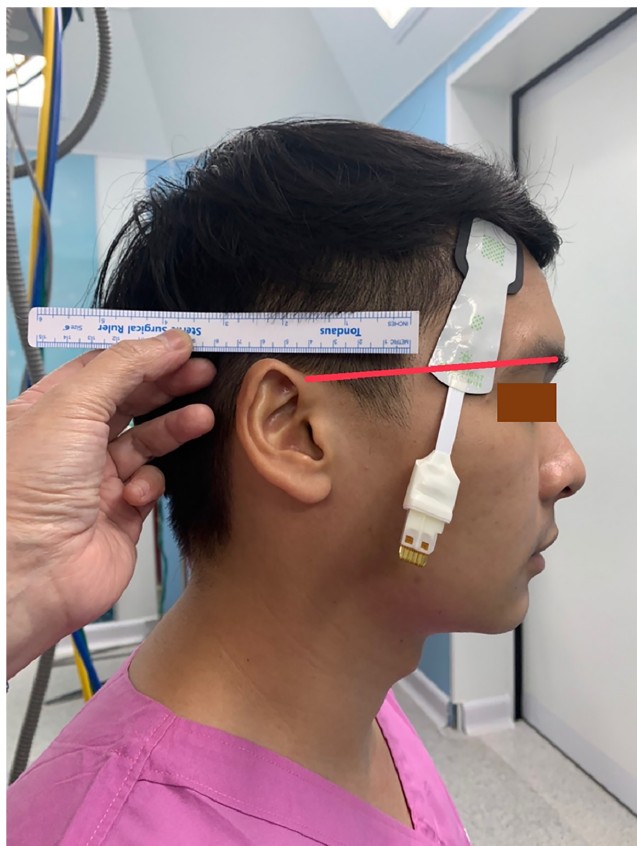

**Figure 2 The location of the temporal sensor.** The proximal light source was located between the eyebrow's tail and the helix root outside the hairline, while the distal end of the deep detector was located upward below the hairline.

adjusted to maintain an end-tidal $CO_2$ close to 35 mmHg. The depth of anesthesia was maintained at one minimum alveolar concentration (MAC) using sevoflurane 1–2% or desflurane 3–6% with 50% $O_2$ and air. The radial artery was cannulated and connected to a

FloTrac transducer (Edwards Lifesciences, Irvine, CA, USA) linked to the HemoSphere monitor to measure invasive blood pressure, stroke volume variation (SVV), and cardiac index (CI). The internal jugular vein was cannulated and connected to another pressure transducer linked to the HemoSphere to measure the systemic vascular resistance index (SVRI). According to the institutional standard protocols, other specific anesthetic management regarding each surgical procedure was accomplished. Patient hemodynamics were optimized using the early goal-directed therapy (EGDT) protocol. The goals were: mean arterial pressure (MAP) 65–90 mmHg; urine output $\geq 0.5$ mL·kg$^{-1}$·h$^{-1}$; SpO$_2$ > 95%; and hematocrit $\geq 30\%$. For patients undergoing cardiac surgery with a cardiopulmonary bypass (CPB), the goals during CPB were MAP 50–75 mmHg, hematocrit 22–25%, PaO$_2$ 150–200 mmHg, and normocapnea. All patients received fluid and blood components to maintain a SVV < 13%, inotropic drugs to achieve a CI of [2.2–4.0] L·min$^{-1}$·m$^{-2}$, followed by vasoactive drugs to maintain a SVRI of 1,500–2,500 dynes·s$^{-1}$·cm$^{-5}$·m$^{-2}$. Arterial blood gas and electrolytes were monitored hourly and corrected as required.

## Statistical analysis

Continuous data were presented as means ± standard deviations (SD) or medians with an interquartile range, as appropriate. Categorical data were presented as numbers (%). Data from A1 *vs.* B1 and A2 *vs.* B2 were assessed for correlation coefficients. The agreements between data from A1 *vs.* B1 and A2 *vs.* B2 were evaluated using the Bland-Altman plot and analysis. The Bland-Altman analysis is a graphical method used to determine whether two methods of measurement can be used interchangeably by plotting the calculated mean difference between the two methods of measurement (the 'bias') with 95% limits of agreement calculated from ±1.96 SD of the mean difference (*Giavarina, 2015*).
All statistical analyses were performed using MedCalc version 20.027.

## RESULTS

Twenty-one patients were recruited between November 1, 2021 and March 10, 2022. There were two drop-outs (ID10 and ID20) due to sensor errors resulting in incomplete data. Therefore, the data (14,364 sets) from 19 patients were analyzed. A summary of the demographic and clinical data for the patients is presented in Table 1. Nearly half the participants were male, and the mean age of all participants was 57.3 years. The types of operation included cardiac, thoracic, video-assisted thoracoscopic, and endovascular stent surgery. The descriptive characteristics of data from A1, B1, A2, and B2 are presented in Table 2. The data from A1 *vs.* B1 and A2 *vs.* B2 show a moderate positive correlation (Table 3).

According to the Bland-Altman analysis, the respective bias of A1 *vs.* B1 and A2 *vs.* B2 was −2.3% (95% CI [−2.5 to −2.2]; $P < 0.0001$) and 0.7% (95% CI [0.6–0.8]; $P < 0.0001$). The respective lower and upper limits of agreement for A1 *vs.* B1 and A2 *vs.* B2 are presented in Table 4. The Bland-Altman plots for A1 *vs.* B1 and A2 *vs.* B2 are presented in Figs. 3 and 4.

Table 1 Demographic and clinical data of the patients ($n$ = 19).

|  | Value |
| --- | --- |
| Male | 10 (52.6) |
| Age (y) | 57.3 ± 16.6 |
| Body weight (kg) | 62.0 ± 10.8 |
| Height (cm) | 165.2 ± 8.0 |
| ASA classification |  |
| II | 9 (47.4) |
| III | 10 (52.6) |
| Underlying diseases |  |
| Hypertension | 7 (36.8) |
| Diabetes mellitus | 2 (10.5) |
| Myocardial ischemia | 3 (15.8) |
| Dyslipidemia | 2 (10.5) |
| Chronic kidney disease | 3 (15.8) |
| Type of operations |  |
| CABG | 3 (15.8) |
| OPCAB | 1 (5.3) |
| Valvular | 3 (15.8) |
| Valvular and aortic root | 2 (10.5) |
| Thoracic | 3 (15.8) |
| VATs | 4 (21.0) |
| Endovascular stent | 2 (10.5) |
| Cone operation | 1 (5.3) |
| Mean arterial pressure (mmHg) | 87.0 ± 16.6 |
| Heart rate (bpm) | 71.3 ± 13.1 |
| $SpO_2$ (%) | 99.0 ± 2.9 |
| End-tidal $CO_2$ | 33.1 ± 4.4 |
| Temperature (°C) | 35.8 ± 0.5 |
| Hemoglobin (g/dL) | 12.3 ± 2.0 |
| P/F ratio (mmHg) | 369.3 ± 121.9 |
| Blood sugar (mg/dL) | 126.8 ± 41.4 |
| Anesthesia time (min) | 314.3 ± 110.9 |

Note:
Data are presented as mean ± SD or number (%).

The results of the Bland-Altman subgroup analysis for sex (male *vs.* female), age (<60 y *vs.* ≥60 y), and anesthetic time (<300 min *vs.* ≥300 min) revealed comparable biases—less than ±5%—and limits of agreements—within ±15% of the biases (Table 5).

## DISCUSSION

The results of the present study showed that the $rScO_2$ values measured from the temporal area slightly but significantly differed from the frontal area. NIRS sensors attached at different areas of the same subject yielded varying $rScO_2$ values. *Kishi et al. (2003)* investigated the effects of sensor location on $rScO_2$ as measured using a cerebral oximeter

**Table 2 Descriptive data of A1, B1, A2, B2 (19 patients).**

|  | A1 | B1 | A2 | B2 |
|---|---|---|---|---|
| N (sample sets) | 14,364 | 14,364 | 14,364 | 14,364 |
| Mean (%) | 71.4 | 73.7 | 72.1 | 71.3 |
| Median (%) | 71.0 | 74.0 | 71.0 | 71.0 |
| Standard deviation (%) | 7.0 | 9.9 | 7.2 | 8.9 |
| Minimum (%) | 48 | 48 | 50 | 47 |
| Maximum (%) | 89 | 98 | 94 | 95 |

**Table 3 Correlation between A1 *vs.* B1 and A2 *vs.*-B2 (19 patients).**

|  | A1-B1 | A2-B2 |
|---|---|---|
| Sample sets | 14,364 | 14,364 |
| Correlation coefficient (r) | 0.627 | 0.548 |
| Significant level | *P* < 0.0001 | *P* < 0.0001 |
| 95% confidence interval for r | [0.617–0.637] | [0.536–0.559] |

**Table 4 Bland-Altman statistics of A1 *vs.* B1 and A2 *vs.* B2 (19 patients).**

|  | A1-B1 (*n* = 14,364) | A2-B2 (*n* = 14,364) |
|---|---|---|
| Bias (%) | −2.3 (95% CI [−2.5 to −2.2]) | 0.7 (95% CI [0.6 to 0.8]) |
| *P* value | <0.0001 | <0.0001 |
| Lower limit of agreement (%) | −17.5 (95% CI [−17.7 to −17.3]) | −14.6 (95% CI [−14.8 to −14.4]) |
| Upper limit of agreement (%) | 12.8 (95% CI [12.6–13.0]) | 16.0 (95% CI [15.8–16.3]) |

INVOS 4100 and applying the sensors to the right forehead (R), 1 cm lateral to R (R1), the left forehead (L), 1 cm lateral to L (L1), and the center of the forehead (C). They found that the $rScO_2$ values from R1 (58 ± 11%) and L1 (59 ± 10%) were significantly lower than those from R (61 ± 10%) and L (61 ± 11%), while the values from C (64 ± 12%) were significantly higher than those at the other sites. *Cho et al. (2017)* revealed that the NIRS sensors located at the upper forehead, compared with the lower forehead, yielded lower $rScO_2$ values. The explanation is that the difference in structures over the brain and the depth of the brain surface affect the optical path length and light scatter. The absolute $rScO_2$ values, thus, may not reflect cerebral oxygenation status; instead, the change in $rScO_2$ values used as a trend monitor has more clinical relevance. Thus, the $rScO_2$ values derived from different scalp areas, with acceptable agreement, may be used for trend monitoring of change in $rScO_2$ values.

The current study revealed that the $rScO_2$ values—as measured by sensors on the frontal and temporal areas—demonstrated moderate correlation with sufficiently good agreement. The biases of data from A1 *vs.* B1 and A2 *vs.* B2 were marginal—less than 3%—with high precision—95% CIs range within 0.3%—indicating excellent agreement and

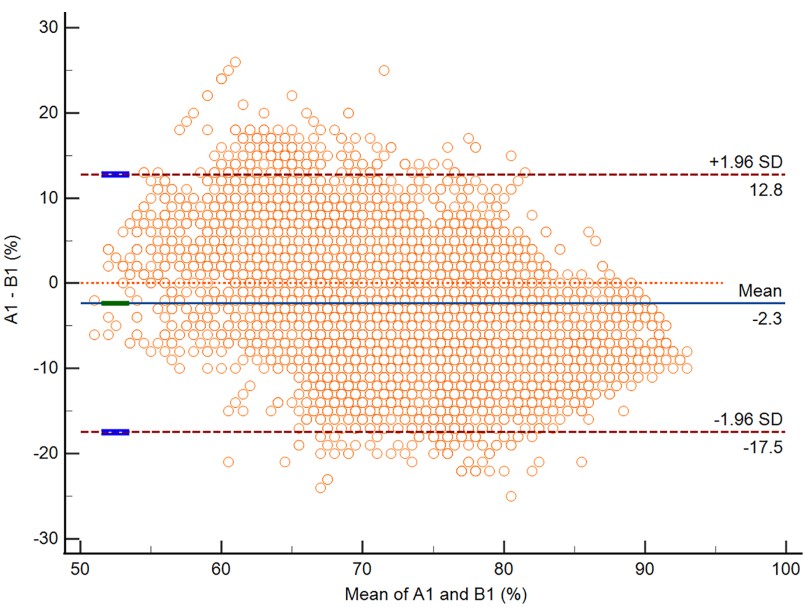

**Figure 3 The Bland-Altman plot of data from A1 *vs*. B1.**

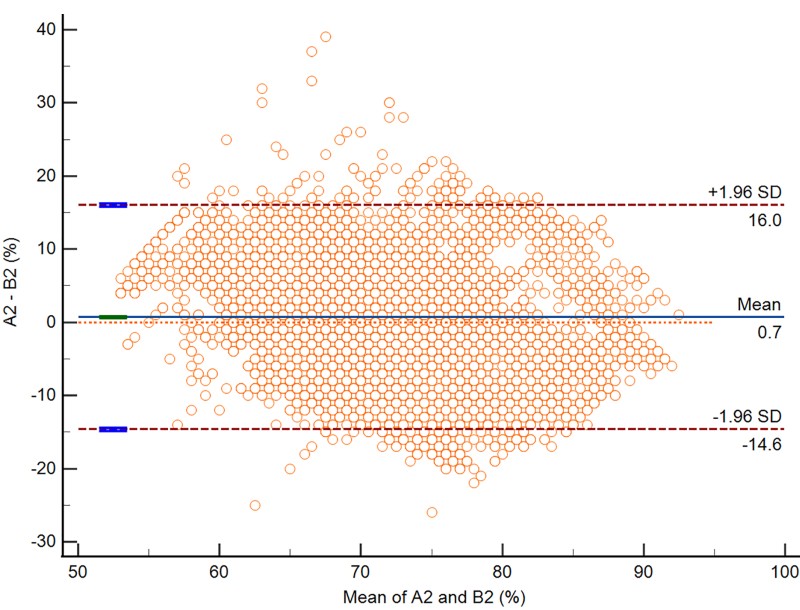

**Figure 4 The Bland-Altman plot of data from A2 *vs*. B2.**

confirming that the temporal area is an alternative site if the NIRS sensors cannot be attached to the forehead. The acceptable lower and upper limits of agreement were defined before the study to be within 15% of the bias because these limits should be narrower than the 20% reduction which is the criteria of cerebral desaturation. The lower and upper limits of agreement of A1 *vs*. B1 and A2 *vs*. B2 were approximately ±15% of the bias, which is clinically acceptable since this margin is within the predefined criteria.

**Table 5  Bland-Altman statistics of A1 *vs.* B1 and A2 *vs.* B2: subgroup analysis.**

| | | A1–B1 | A2–B2 |
|---|---|---|---|
| Sex: Male (*n* = 5,686) | Bias (%) | −1.8 (95% CI [−2.0 to −1.7]) | 4.7 (95% CI [4.5–4.9]) |
| | *P* value | <0.0001 | <0.0001 |
| | Lower limit of agreement (%) | −15.0 (95% CI [−15.3 to −14.7]) | −7.9 (95% CI [−8.2 to −7.6]) |
| | Upper limit of agreement (%) | 11.3 (95% CI [11.0–11.6]) | 17.3 (95% CI [17.0–17.6]) |
| Sex: Female (*n* = 8,678) | Bias (%) | −2.7 (95% CI [−2.8 to −2.5]) | −1.9 (95% CI [−2.0 to −1.7]) |
| | *P* value | <0.0001 | <0.0001 |
| | Lower limit of agreement (%) | −17.5 (95% CI [−17.8 to −17.2]) | −16.9 (95% CI [−17.1 to −16.6]) |
| | Upper limit of agreement (%) | 12.3 (95% CI [12.0–12.6]) | 13.1 (95% CI [12.8–13.4]) |
| Age: <60 y (*n* = 7,509) | Bias (%) | −0.2 (95% CI [−0.4 to −0.0]) | 4.9 (95% CI [4.8 to −5.1]) |
| | *P* value | 0.0338 | <0.0001 |
| | Lower limit of agreement (%) | −15.1 (95% CI [−15.4 to −14.8]) | −8.1 (95% CI [−8.4 to −7.9]) |
| | Upper limit of agreement (%) | 14.1 (95% CI [13.8–14.4]) | 18.0 (95% CI [17.7 to 18.2]) |
| Age: ≥60 y (*n* = 6,855) | Bias (%) | −4.7 (95% CI [−4.8 to −4.5]) | −3.9 (95% CI [−4.0 to −3.7]) |
| | *P* value | <0.0001 | <0.0001 |
| | Lower limit of agreement (%) | −18.3 (95% CI [−18.6 to −18.0]) | −16.4 (95% CI [−16.6 to −16.1]) |
| | Upper limit of agreement (%) | 8.9 (95% CI [8.7–9.2]) | 8.6 (95% CI [8.4–8.9]) |
| Anesthetic time: <300 min (*n* = 3,648) | Bias (%) | −1.2 (95% CI [−1.4 to −1.0]) | 1.8 (95% CI [1.5–2.0]) |
| | *P* value | <0.0001 | <0.0001 |
| | Lower limit of agreement (%) | −14.7 (95% CI [−15.1 to −14.3]) | −11.4 (95% CI [−11.8 to −11.0]) |
| | Upper limit of agreement (%) | 12.4 (95% CI [12.0–12.7]) | 14.9 (95% CI [14.5–15.3]) |
| Anesthetic time: ≥300 min (*n* = 10,716) | Bias (%) | −2.7 (95% CI [−2.9 to −2.6]) | 0.4 (95% CI [0.2–0.5]) |
| | *P* value | <0.0001 | <0.0001 |
| | Lower limit of agreement (%) | −17.3 (95% CI [−17.6 to −17.1]) | −14.5 (95% CI [−14.8 to −14.2]) |
| | Upper limit of agreement (%) | 11.9 (95% CI [11.6–12.2]) | 15.4 (95% CI [15.2–15.7]) |

The data from two patients (ID10 and ID20) were not included in the final analysis due to sensor error. After confirming the correctness of the sensor attachment, the HemoSphere monitor continued to report incomplete data.

Since many factors may affect $rScO_2$ values, *e.g.*, sex, age, and anesthetic time (*Kishi et al., 2003*; *Chan et al., 2017*), a subgroup analysis was performed and found that these factors did not affect the reading of $rScO_2$ values in the current study. The biases and limits of agreements of these subgroups were comparable to the primary outcomes. The results of

the current study differ from those of *Kishi et al. (2003)* who found a significant negative correlation between age and $rScO_2$ values. This may be due to differences in the age range of the subjects (7–89 y in the study of Kishi et al. *vs.* 21–78 y in the current study).

The explanation for the agreement of $rScO_2$ values reading from the frontal and temporal areas is that both the frontal and temporal lobes are parts of the cerebral cortex with common blood supplies and covering structures, thus the $rScO_2$ values measured from these areas should be comparable. Even though the differences were statistically significant, they are so trivial that there is no clinical relevance. Since the baseline $rScO_2$ values in cardiac surgery have high variability (*Chan et al., 2017*), the trend in values is more clinically relevant (*Vegh, 2016*).

Many factors affect $rScO_2$ value including arterial carbon dioxide, cardiac output, arterial blood pressure, hemoglobin concentration, neural excitation, and anesthetic depth (*Vegh, 2016*). These factors should be optimized and taken into account to correct cerebral desaturation. Skin color does not affect the $rScO_2$ reading (*Vegh, 2016*). Scalp hair follicles (SHF) have strong impact on NIRS measurement by decreasing the detected light intensity signal by 15–80% resulting in a miscalculation of $rScO_2$ by 11.7% to 292.2% linearly at SHF density varied from 1% to 11% in Asian human (*Fang et al., 2018*). The authors tried to attach the NIRS sensors to the shaved occiput and the temporal area including the hairline and found no $rScO_2$ reading on the HemoSphere monitor. Thus, the NIRS sensor should not be attached to the scalp area where there are hair follicles.

POCD is a permanent drop in cognitive function after surgery that interferes with quality of life of the patients. Cerebral desaturation was identified to be associated with POCD (*Casati et al., 2005*). The significant risk factors for POCD are increasing age and maximum percentage drop in $rScO_2 > 11\%$ (*Lin et al., 2013*). A network meta-analysis showed that maintaining $rScO_2 > 80\%$ of the baseline value was protective against POCD/POD in cardiac surgery patients (*Ortega-Loubon et al., 2019*). Thus, $rScO_2$ monitoring for cerebral oxygenation during cardiac surgery is essential. The standard site for NIRS sensor attachment is the forehead, but an alternative site is needed in case the forehead is not available.

The temporal area was chosen as an alternative to the forehead because this is the only scalp area that has no hair follicles with similar depth from scalp to brain tissue. Furthermore, these two areas reflect the frontal and temporal lobes, which form the cerebral cortex with a common blood supply (*El-Baba & Schury, 2022*; *Patel, Biso & Fowler, 2022*). The near-infrared spectrum of NIRS can penetrate the tissue up to 2.5 cm (*Steppan & Hogue, 2014*); thus, the sensors attached to these areas can accurately measure the oxygen saturation of the underlying brain tissue resulting in comparable $rScO_2$ values.

The Bland-Altman analysis was selected to compare the $rScO_2$ values measured from sensors attached to the frontal and temporal areas because this method quantifies agreement between two quantitative measurements by assessing the bias between the mean differences and constructing limits of agreement (*Giavarina, 2015*). The method does not judge whether those biases and limits are acceptable or not. Acceptable limits must be defined before the study based on clinical essentiality. The biases and limits of agreement

of this study remain within the proposed criteria, indicating that the temporal area may be used as a site for attaching NIRS sensors if the frontal area is not feasible.

### Limitations

A limited number of patients undergoing elective cardiothoracic or vascular surgery at a single tertiary center were included, so the results may not be generalizable to other contexts. In addition, this study used the ForeSight Elite sensors and HemoSphere monitoring that again may not be generalizable to other types of monitors. Therefore, a further multi-center study with a larger sample size with more types of monitors is recommended to validate our findings.

## CONCLUSIONS

The $rScO_2$ values measured from the sensors attached to the frontal and temporal areas showed moderate correlation. The biases and limits of agreement remained within the predefined criteria indicating a good agreement. The temporal area may thus be used as an alternative site to the frontal area for cerebral oximetry monitoring. The identification of cerebral desaturation should be based on the trend—in changes to $rScO_2$ values from baseline—rather than the absolute $rScO_2$ values.

## ACKNOWLEDGEMENTS

The authors thank Mr. Bryan Roderick Hamman under the aegis of the Publication Clinic, Khon Kaen University, Thailand, for assistance with the English-language presentation of the manuscript.

### Funding

The study was funded by an unrestricted University grant from the Faculty of Medicine, Khon Kaen University, Khon Kaen, Thailand (Grant number: IN65103) and supported by the Cardiovascular and Thoracic Surgery Research Group, Khon Kaen University.
The funders had no role in study design, data collection and analysis, decision to publish, or preparation of the manuscript.

### Grant Disclosures

The following grant information was disclosed by the authors:
Faculty of Medicine, Khon Kaen University: IN65103.
Cardiovascular and Thoracic Surgery Research Group, Khon Kaen University.

### Competing Interests

The authors declare that they have no competing interests.

## Author Contributions

- Sirirat Tribuddharat conceived and designed the experiments, performed the experiments, analyzed the data, prepared figures and/or tables, authored or reviewed drafts of the article, and approved the final draft.
- Kriangsak Ngamsaengsirisup conceived and designed the experiments, performed the experiments, analyzed the data, prepared figures and/or tables, authored or reviewed drafts of the article, and approved the final draft.
- Phatcharakamon Mahothorn conceived and designed the experiments, performed the experiments, analyzed the data, prepared figures and/or tables, authored or reviewed drafts of the article, and approved the final draft.
- Thepakorn Sathitkarnmanee conceived and designed the experiments, performed the experiments, analyzed the data, prepared figures and/or tables, authored or reviewed drafts of the article, and approved the final draft.

## Human Ethics

The following information was supplied relating to ethical approvals (*i.e.*, approving body and any reference numbers):

The Khon Kaen University Ethics Committee for Human Research approved the study (HE641311).

## Data Availability

The raw data is available in the Supplemental Files.

## Supplemental Information

Supplemental information for this article can be found online at http://dx.doi.org/10.7717/peerj.14058#supplemental-information.

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
