# Peer review of "Correlation and agreement of regional cerebral oxygen saturation measured from sensor sites at frontal and temporal areas in adult patients undergoing cardiovascular anesthesia"

_PeerJ, doi:10.7717/peerj.14058_

## Round 0.1 · original submission · Major Revisions

I read your manuscript, and I feel it is needed for this research field although the manuscript will be needed some revisions. Then, I received three reports from three reviewers. They also feel this manuscript is needed for your research field but it is needed to improve for the publication in PeerJ. Please find the attached reports from the reviewers.

Reviewer 1 ·

Basic reporting

Tribuddharat et al. present a manuscript entitled " Correlation and agreement of regional cerebral oxygen saturation measured from sensor sites at frontal and temporal areas in adult patients undergoing cardiovascular anaesthesia" which aims to find an alternative region of the brain for using NIRS. While the need for finding and alternative region for placement of NIRS sensors is apparent, the study needs significant work.

Experimental design

As the authors mention in the manuscript, the conducted clinical study is very small with limited demographic variability. With measurement techniques such as NIRS, characteristics like physiological condition, skin pigmentation etc. play a major role in determining the sensor validity. Even if the dataset can not be increased to a high number of patients, some variability in the demographics will help understand the correlation better.

Validity of the findings

1. The authors show some correlation between the sensor readings from the frontal and temporal lobe. However, the correlation of the data to physiological relevance still needs to be addressed.
• For instance, Slone et al. reported that neuronal activity, occurs in the ipsilateral frontal lobe prior to the onset of temporal lobe seizures. (Ref: Slone, Edward, et al. "Near-infrared spectroscopy shows preictal haemodynamic changes in temporal lobe epilepsy." Epileptic Disorders 14.4 (2012): 371-378). Thus, if the target condition requires the sensors to detect an early onset frontal lobe would prove to be a better measurement site.

• Tribuddharat et al. mention in the manuscript that NRIs measurements can help lower the risks of postoperative stroke, postoperative cognitive dysfunction (POCD), and postoperative delirium (POD). It would be helpful for them to assess and discuss the medical aspect of measuring oxygenation from the temporal region rather than the frontal lobe.

• Another point of debate is the presence of interfering biological molecules in the temporal region which may share same optical characteristics with haemoglobin such as myoglobin. Such molecules may become a major contributor confounding the sensor results. Therefore, the medical relevance really needs to be addressed clearly, to ensure that measurement from the temporal lobe correlate with results from the frontal lobe.

2. The authors report a correlation coefficient of 0.627 and 0.548 for the A1-B1 and A2-B2 sets respectively. The authors use the term strong correlation multiple times in the manuscript, however from a medical device perspective the values imply a low to moderate correlation (Ref: Mukaka, Mavuto M. "A guide to appropriate use of correlation coefficient in medical research." Malawi medical journal 24.3 (2012): 69-71). This issue should be addressed.

Reviewer 2 ·

Basic reporting

Thanks for the opportunity to provide a peer review for this interesting article. This manuscript discusses a comparison between sensor sites at frontal and temporal areas to measure regional cerebral oxygen saturation (rScO2). By doing a Bland-Altman Analysis between the two measurement records, the authors conclude that the temporal area may be an alternative site for cerebral oximetry monitoring. Overall, the language used throughout the whole text is sufficient. For literature references, the reviewer prefers to see more references for the background of cerebral oximetry monitoring. For example, the following citation could be a good reference for monitoring cerebral oxygen saturation under anesthesia: Végh T. (2016). Cerebral Oximetry in General Anaesthesia. Turkish journal of anaesthesiology and reanimation, 44(5), 247–249. https://doi.org/10.5152/TJAR.2016.26092016. In addition, more reference could be provided to support why choosing temporal area as the alternative site for monitoring in line 71-72.

Experimental design

The authors did a good job stating the criteria on how to choose the patients for this set of assessments, and the research question is defined and quite straightforward. However, there are a few points worthy of discussion:
1. To ensure the consistency of the recording, a more defined location where the sensors are attached could be provided. For example, how many centimeters away from the ear?
2. Are all the patients having a comparable duration of their operations? The duration of anesthesia could also affect the reading of cerebral oxygen level. A statistical analysis of this parameter could provide more confidence in the conclusion.
3. Since the statistics are critical for the conclusion, a more specific explanation of the Bland-Altman Analysis could be elaborated on in the method section.

Validity of the findings

The reviewer has the following concerns regarding the results:
1. For table 1, please consider using n=19, since ID 10 and ID 20 are not included for Bland-Altman analysis in Figures 2 & 3.
2. Why do IDs 17,18 and 20 not indicate FiO2 in the summarized data sheet?
3. Please consider increasing the n number.

·

Basic reporting

General comments
This study is of importance to neurological science as it is seemingly contributory to improving fundamental understanding of the working schematics of the human brain. However, several technical and major issues need to be addressed for the standardization of the manuscript.
The background has not declared the motivation for the study, only the objective was mentioned. In simpler terms, WHY was the study conducted? This should be succinctly stated in the background section of the abstract

The use of possessive pronouns such as “we” and “our”, for example “159 We chose the temporal area as an alternative to… 164 We selected the Bland-Altman analysis…” is not too ethical for technical writing and should be revised for the entire manuscript. The above example can be restructured as “159 The temporal area was chosen as an alternative to… 164 The Bland-Altman analysis was selected…”

Graphical abstract
A graphical abstract showing a figurative expression of the patients head, spectrogram, and other important details would greatly suffice. The authors should strongly consider adding this to the manuscript.

Experimental design

The methods section, (both of the abstract and main body) should be more detailed on the settings used for each sensor, if such exists. This would strongly improve the originality of the project work.

Results
The result section states that “129 Twenty-one patients were recruited between November 1, 2021 and March 10, 2022. 130 There were two drop-outs (ID10 and ID20) due to sensor errors resulting in incomplete data. 131 The data (14,364 sets) from 19 patients were…” This sounds more like a portion of the materials and methods. The result is expected to be an exploratory declaration of the output obtained from the data. It will be beneficial if this section of the manuscript is completely revised and restructured appropriately.

Validity of the findings

Discussion
There are few issues with this section of the manuscript. To begin with, only one study was effectively referenced (Kishi et al) and it was even referenced without its reference number. Furthermore, the discussion section seems like what should be in the result section: it is entirely a declaration of results, and nothing more. No discussions on WHY the result trends were obtained, with multiple references. No discussion on what could have caused the results (with multiple references): maybe the brain tissues? maybe the gender of the subjects? Maybe the age of the subjects? Nothing was mentioned. This section should be rewritten.
Conclusions
Only ONE conclusion was made, and as such this portion should not be termed as a conclusion. More work should be done on this manuscript. The conclusions should arise from the output of every variable taken into consideration and studied in the course of the experimentation. Furthermore, there should be recommendation for subsequent works related to this subject.

---

## Round 0.2 · Major Revisions

I received the review comments from our reviewers. Please find the review comments from the reviewers. Especially, one reviewer would feel some critical questions were not addressed adequately.

Reviewer 2 ·

Basic reporting

Great appreciation to the authors for their diligent efforts to respond to the previous comments on the manuscript. The reviewer has carefully reviewed the revised paper, and feels some critical questions were not addressed adequately. The following comments have been made:

The critical concern is the lack of clinical relevance for the alternative sensor location. Given the various types of surgical patients involved and n number in this study, choosing this location mainly based on the blood supply and anatomical structures is very thinly supported. There are quite a few papers reporting drastic value changes between small variables, e.g the comparison between sensor locations of one centimeter apart (Kishi K, Kawaguchi M, Yoshitani K, Nagahata T, Furuya H. J Neurosurg Anesthesiol. 2003).

1. Previous comments regarding citations for cerebral oximetry monitoring:
Though review articles are appropriately used as overview citations for broad scientific topics or ideas, most citations, especially those focusing on previously published concepts or results, should be of original research papers. Research and review from multiple sources could provide a more unbiased view on the topic .

2. Clinical relevance on choosing the temporal site:
The reviewer understands that the literature for using temporal site as a sensor location may be minimum, which could indicate the significance of existence of this study. But choosing temporal site only based on the blood supply and anatomical structure is not sufficient, especially when this study includes various types of surgeries (e.g. lung lobectomy, abdominal endovascular aneurysm repair, thymectomy etc.), which could include quite different complications. The clinical relevance is definitely worthy of proving in depth.

On the same topic, reviewer #1 actually pointed out a very good question as followed:
"The authors show some correlation between the sensor readings from the frontal and temporal lobe. However, the correlation of the data to physiological relevance still needs to be addressed.
• For instance, Slone et al. reported that neuronal activity, occurs in the ipsilateral frontal lobe prior to the onset of temporal lobe seizures. (Ref: Slone, Edward, et al. "Near-infrared spectroscopy shows preictal hemodynamic changes in temporal lobe epilepsy." Epileptic Disorders 14.4 (2012): 371-378). Thus, if the target condition requires the sensors to detect an early onset frontal lobe would prove to be a better measurement site."

The authors have not adequately answered commenter #1's comments. The key question is what does this measurement rScO2 mean in the context of surgery? Does it correlate with a specific type of disease? Different types of disease or surgery may show different patterns of neuronal activity, resulting in different readings of oxygen levels (rScO2). Commenter #1 gave an example of some neuronal activity that occurs only in the frontal lobe, not the temporal lobe. In this case, the alternative site would not work. The authors still have not been able to point out what the clinical significance is behind measuring rScO2 in the temporal lobe. It would be more convincing to show if there is a similarity in terms of the risks or complications involved in all the types of procedures included in this data set so that measuring the frontal and temporal lobes would give the same results, rather than simply stating that they both come from the same vessel.

Experimental design

The authors address some of the previous comments on the experimental design. For the Bland-Altman analysis. Yes, the reviewer can see that the confidence interval for A1 vs. B1 is 95% at -2.5 to -2.2 and the confidence interval for A2 vs. B2 is 95% at 0.6 to 0.8. How is this clinically interpreted? Why is the CI range wider for position 1 than for position 2? Is this difference large enough to matter? This is a clinical question, not a statistical one. At the very least, the significance of this difference can be discussed.

Validity of the findings

1. Two patients in this study had incomplete data due to sensor errors, was this due to sensor placement? This needs to be discussed at least in the discussion section.

Since the manuscript talks about the feasibility of using alternative sites for sensor placement, whether the location of the sensor provides consistent value should also be discussed. The authors mention that two patients were excluded due to sensor error. What was the sensor error? Was the sensor prone to disconnection because of the new location? At least in the discussion, this question is worth addressing.

2. Is the current number of n sufficient? sample size tested?

The authors believe that 14,364 data points from 19 patients are sufficient. The reviewer thinks that using data points rather than the number of patients as the n number would lead to the following concerns. When looking at the data, rScO2 was measured every 20 seconds, which means that the longer the procedure, the more data points there will be for this patient. For example, by simple calculation, patient ID #3 contributed 10.7% of the data points, while patient #15 contributed only about 1.43%. Therefore, the duration of surgery was assumed here as a factor that would not affect the results. However, all these variables were not excluded as potentially affecting the readings, which is supported by various literature and needs to be taken into account when comparing data points (Cho, AR., Kwon, JY., Kim, C. et al. Effect of sensor location on regional cerebral oxygen saturation measured by INVOS 5100 in on-pump cardiac surgery. J Anesth 31, 178–184 (2017).).

The reviewer suggested the following:
The first suggestion is when doing a Bland-Altman plot, compare the data points after grouping them according to some variables: time of surgery, age, and gender. This comparison will come to some conclusions: either these variables do not affect the original conclusion based on the comparison and they remain comparable, or these classifications may help you to exclude more outliers and thus increase the correlation between the two methods. These data will increase the confidence of the conclusion by showing that these variables were indeed taken into account.
A second suggestion is that you do a sample size test based on the pilot study, which will produce a relative sample size figure that will convince reviewers that the provided data are sufficient to draw conclusions. An example of this can be seen in the methods section of this paper, which was also conducted for the study comparing sensor location and rScO2 measurements.

Additional comments

Overall, this study contains valuable data but needs better organization and interpretation, especially when no sufficient literature background is supported, a conclusion like this must be carefully drawn, after considering all possible variables. The reviewer still thinks this manuscript needs significant work.

·

Basic reporting

The revised manuscript shows quite a number of desirable improvements. The originality of the research has been well enhance. Nonetheless, some minor revisions are suggested.

Experimental design

The experimental design has been presented in a better fashion with the figures. Authors should only include any official permissions as required by ethical and publishing guidelines.

Validity of the findings

Every study is an addition to knowledge. The conclusion aspect of every study should bear some future prospects. It is strongly suggested that authors incorporate this to the conclusion section.

---

## Round 0.3 · accepted · Accept

I am very happy to inform you that your manuscript has been accepted for publication.

·

Basic reporting

Authors have answered all demanded revisions and significantly improve the technicality of the manuscript. Its acceptance is suggested.

Experimental design

It is seemingly satisfactory.

Validity of the findings

It is seemingly satisfactory.